# Balanced networks under spike-time dependent plasticity

**Alan Eric Akil** [1], **Robert Rosenbaum** [2,3], **Krešimir Josić** [1,4]*

**1** Department of Mathematics, University of Houston, Houston, Texas, United States of America,
**2** Department of Applied and Computational Mathematics and Statistics, University of Notre Dame, Notre Dame, Indiana, United States of America, **3** Interdisciplinary Center for Network Science and Applications, University of Notre Dame, Notre Dame, Indiana, United States of America, **4** Department of Biology and Biochemistry, University of Houston, Houston, Texas, United States of America

* josic@math.uh.edu

**Data Availability Statement:** There is no data per se, but all code is available at https://github.com/alanakil/PlasticBalancedNetsPackage.

**Funding:** Funding was provided by grants NIH-1R01MH115557 (KJ), NSF DMS-1654268 (RR) and DBI-1707400 465 (AA, RR, and KJ). National

## Abstract

The dynamics of local cortical networks are irregular, but correlated. Dynamic excitatory–inhibitory balance is a plausible mechanism that generates such irregular activity, but it remains unclear how balance is achieved and maintained in plastic neural networks. In particular, it is not fully understood how plasticity induced changes in the network affect balance, and in turn, how correlated, balanced activity impacts learning. How do the dynamics of balanced networks change under different plasticity rules? How does correlated spiking activity in recurrent networks change the evolution of weights, their eventual magnitude, and structure across the network? To address these questions, we develop a theory of spike–timing dependent plasticity in balanced networks. We show that balance can be attained and maintained under plasticity–induced weight changes. We find that correlations in the input mildly affect the evolution of synaptic weights. Under certain plasticity rules, we find an emergence of correlations between firing rates and synaptic weights. Under these rules, synaptic weights converge to a stable manifold in weight space with their final configuration dependent on the initial state of the network. Lastly, we show that our framework can also describe the dynamics of plastic balanced networks when subsets of neurons receive targeted optogenetic input.

## Author Summary

Animals are able to learn complex tasks through changes in individual synapses between cells. Such changes lead to the coevolution of neural activity patterns and the structure of neural connectivity, but the consequences of these interactions are not fully understood. We consider plasticity in model neural networks which achieve an average balance between the excitatory and inhibitory synaptic inputs to different cells, and display cortical–like, irregular activity. We extend the theory of balanced networks to account for synaptic plasticity and show which rules can maintain balance, and which will drive the network into a different state. This theory of plasticity can provide insights into the relationship between stimuli, network dynamics, and synaptic circuitry.

Institutes of Health - https://www.nih.gov National Science Foundation - https://nsf.gov. The funders had no role in study design, data collection and analysis, decision to publish, or preparation of the manuscript.

**Competing interests:** The authors have declared that no competing interests exist.

## Introduction

Cortical neuronal activity is irregular, correlated, dominated by a low dimensional component [1–6], and characterized by a balance between excitation and inhibition [7–12]. Such balance is now widely thought to give rise to stable, irregular neural activity [1, 7, 12–18]. Early theoretical work has focused on irregular asynchronous dynamics, with large networks exhibiting vanishing correlations [13, 19]. However, more recent extensions have shown how correlated dynamics can be generated both endogenously and exogenously, while preserving irregular single cell activity [20–28], showing the existence of both asynchronous and correlated states in balanced networks.

Correlated firing can also produce changes in synaptic weights [29, 30]. For instance spike–time dependent plasticity (STDP), is driven by patterns in the timing of pre– and post–synaptic spikes [31, 32]. However, we still lack a theory that relates STDP to changes in neural activity, and the resulting neural computations. Hence, often the analysis of the effects of STDP relies on simulations [29, 33–35]. Analytical treatments have been proposed for a number of cases, starting with the description of mean synaptic dynamics of a single integrate–and–fire neuron receiving feed–forward input from a collection of Poisson neurons [36]. These results have been extended to small networks [34], and networks of Poisson neurons [37–40]. Other work provided analytical treatments of specific plasticity rules, such as homeostatic inhibitory plasticity [41, 42]. Using linear response and motif resumming techniques [43], Ocker et al. developed a theory describing the evolution of mean weights in recurrent neural networks of noisy integrate–and–fire neurons under STDP [44]. This approach relies on the assumption that the input to individual cells is dominated by white noise, local synaptic input is weak, and that the integral of the STDP function is small. Related results were obtained by treating neural firing as a Poisson process [37–39, 45]. In particular, Ravid Tannenbaum et al. showed that in networks of Poisson neurons synfire chains and self connected assemblies can emerge autonomously in recurrent networks [46]. Montangie et al. showed that a more realistic form of STDP based on spike triplets also leads to autonomous emergence of assemblies [47].

Here, we develop a complementary theory describing the evolution of synaptic weights and associated mean rates in tightly balanced networks in both correlated and asynchronous states. We combine the mean–field theory of firing rates and correlations in balanced networks [13, 14, 23, 24, 48–50] with an averaging approach assuming a separation of timescales between changes in spiking activity, and the evolution of synaptic weights [30]. We show how the weights and the network dynamics co–evolve under different classical rules, such as Hebbian plasticity, Kohonen's rule, and a form of inhibitory plasticity [31, 32, 41, 51, 52]. In general, the predictions of our theory agree well with empirical simulations. We also explain when the mean–field theory fails, leading to disagreements with simulations, and we develop a semi–analytic extension of the theory that explains these disagreements.

We find that spike train correlations, in general, have a mild effect on the synaptic weights and firing rates, in agreement with previous work [44, 53]. We also show that for some STDP rules, synaptic competition can introduce correlations between synaptic weights and firing rates, resulting in the formation of a stable manifold of fixed points in weight space, and hence asymptotic weight distributions that depend on the initial state. Finally, we apply this theory to show how inhibitory STDP [41] can lead to a reestablishment of an asynchronous, balanced state that is broken by optogenetic stimulation of a neuronal subpopulation [54]. We thus extend the classical theory of balanced networks to understand how synaptic plasticity shapes their dynamics.

## Materials and methods

### Review of mean–field theory in balanced networks

In mammals, local cortical networks can be comprised of thousands of cells, with each neuron receiving thousands of inputs from cells within the local network, and other cortical layers, areas, and thalamus [55]. Predominantly excitatory, long–range inputs would lead to high, regular firing unless counteracted by local inhibition. To reproduce the sparse, irregular activity observed in cortex, model networks often exhibit a balance between excitatory and inhibitory inputs [13, 14, 19, 23, 48, 56–58]. This balance can be achieved robustly and without tuning, when synaptic weights are scaled like $\mathcal{O}(1/\sqrt{N})$, where $N$ is the network size [13, 14]. In this balanced state mean excitatory and inhibitory inputs cancel one another, and the activity is asynchronous [19]. Inhibitory inputs can also track excitation at the level of cell pairs, cancelling each other in time, and produce a correlated state [1, 23].

We first review the mean–field description of asynchronous and correlated states in balanced networks, and provide expressions for firing rates and spike count covariances averaged over subpopulations that accurately describe networks of more than a few thousand neurons [13, 14, 23, 24, 48–50]: Let $N$ be the total number of neurons in a recurrent network composed of $N_e$ excitatory and $N_i$ inhibitory neurons. Cells in this recurrent network also receive input from $N_x$ external Poisson neurons firing at rate $r_x$, and with pairwise correlation $c_x$ (See Fig 1A, and S1 Appendix for more details). We assume that $q_b = N_b/N \sim \mathcal{O}(1)$ for $b =$ e, i, x. Let $p_{ab}$ be the probability of a synaptic connection, and $j_{ab} \sim \mathcal{O}(1)$ the weight of a synaptic connection from a neuron in population $b =$ e, i, x to a neuron in population $a =$ e, i. For simplicity we assume that both the probabilities, $p_{ab}$, and weights, $j_{ab} \sim \mathcal{O}(1)$, are constant across pairs of subpopulations.

We define the recurrent, and feedforward mean–field connectivity matrices as

$$\overline{W} = \begin{bmatrix} \overline{w}_{ee} & \overline{w}_{ei} \\ \overline{w}_{ie} & \overline{w}_{ii} \end{bmatrix}, \qquad \text{and} \qquad \overline{W}_x = \begin{bmatrix} \overline{w}_{ex} \\ \overline{w}_{ix} \end{bmatrix}, \tag{1}$$

where $\overline{w}_{ab} = p_{ab}j_{ab}q_b \sim \mathcal{O}(1)$.

Let $r = [r_e, r_i]^T$ be the vector of mean excitatory and inhibitory firing rates. The mean external input and recurrent input to a cell are then $\overline{X} = \sqrt{N}\overline{W}_x r_x$ and $\overline{R} = \sqrt{N}\overline{W}r$, respectively, and the mean total synaptic input to any neuron is given by

$$\overline{I} = \sqrt{N}[\overline{W}r + \overline{W}_x r_x]. \tag{2}$$

We next make the ansatz that in the balanced state the mean input and firing rates remain finite as the network grows, *i.e.*, $\overline{I}, r \sim \mathcal{O}(1)$ [13, 14, 23, 48–50]. This is only achieved when external and recurrent synaptic inputs are in balance, that is when

$$\lim_{N\to\infty} r = -\overline{W}^{-1}\overline{W}_x r_x \tag{3}$$

provided that also $\overline{X}_e/\overline{X}_i > \overline{w}_{ei}/\overline{w}_{ii} > \overline{w}_{ee}/\overline{w}_{ie}$ [13, 14]. Eq (3) holds in both the asynchronous and correlated states.

We define the mean spike count covariance matrix as:

$$C = \begin{bmatrix} C_{ee} & C_{ei} \\ C_{ie} & C_{ii} \end{bmatrix} \tag{4}$$

where $C_{ab}$ is the mean spike count covariance between neurons in populations $a =$ e, i and

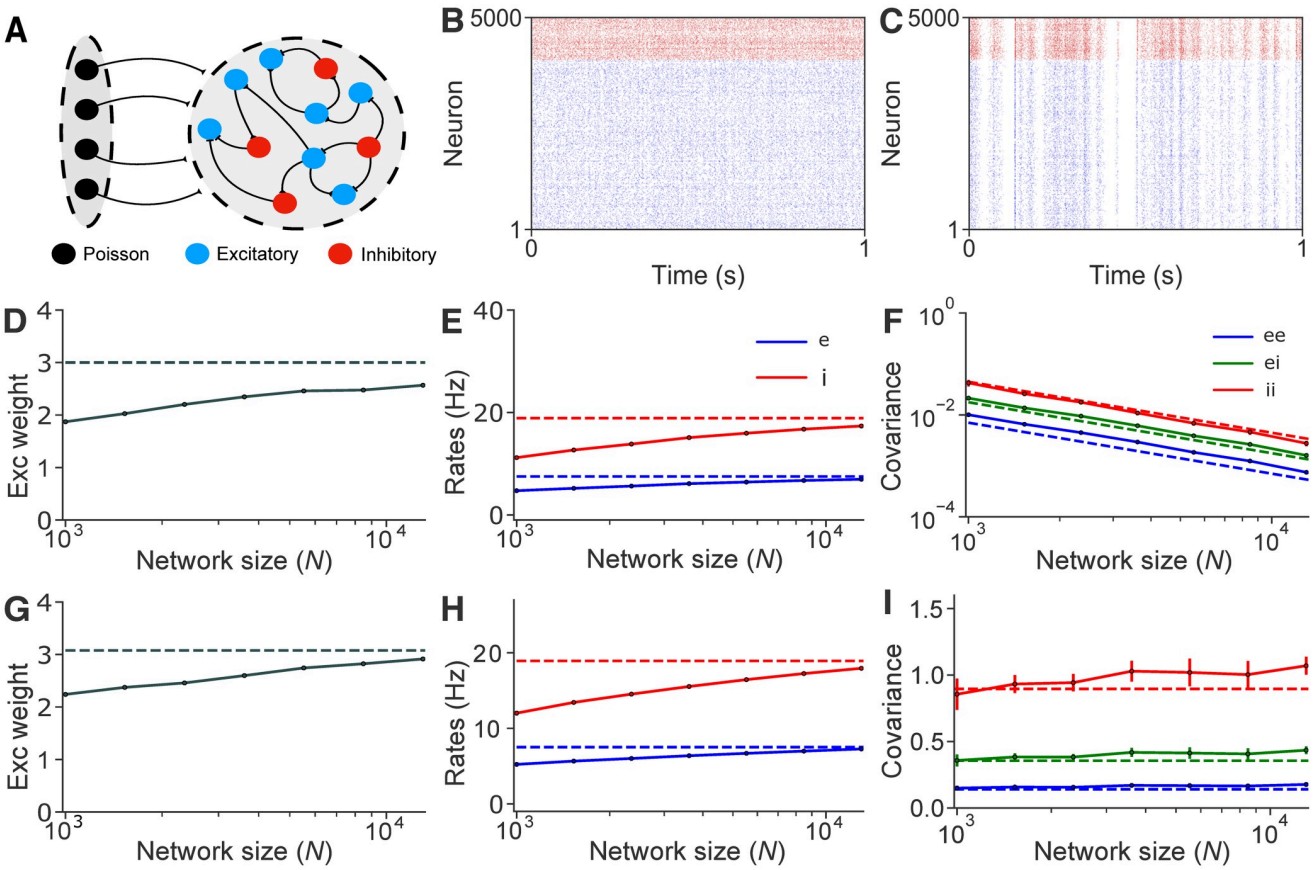

**Fig 1. A plastic, balanced network in asynchronous and correlated regimes. A**: A recurrent network of excitatory, *E*, and inhibitory, *I*, neurons is driven by an external feedforward layer, *X*, of correlated Poisson neurons. **B**: Raster plot of all neurons in a network of *N* = 5000 neurons in an asynchronous state. *E* cells in blue, *I* neurons in red. **C**: Same as (**B**), but in a correlated state. **D**: Mean steady state *EE* synaptic weight, $j_{ee}$, in an asynchronous state. **E**: Mean *E* and *I* firing rates for different network sizes, *N*, in an asynchronous state. **F**: Mean *EE*, *II* and *EI* spike count covariances in an asynchronous state. **G**–**I**: Same as (**D**–**F**) but for a network in a correlated state. Solid lines represent simulations, and dashed lines are values obtained using Eqs (3), (5) and (21). All empirical results were averaged over 10 realizations. In the asynchronous state $c_x = 0$, and in the correlated state $c_x = 0.1$. Unless otherwise stated, colors carry the same meaning in all figures.

*b* = e, i, respectively, counted over time windows of size $T_{\mathrm{win}}$. Throughout all simulations and theoretical predictions, we set $T_{\mathrm{win}} = 250$ ms, however the theory is flexible to other time window sizes.

From [23, 24] it follows that in large networks, to leading order in $1/N$ (See [19, 59–61] for similar expressions derived for similar models),

$$C \approx \frac{1}{N} T_{\mathrm{win}} \overline{W}^{-1} \Gamma \overline{W}^{-T} - \frac{1}{N} \begin{bmatrix} \dfrac{r_e T_{\mathrm{win}} F_e}{q_e} & 0 \\[2mm] 0 & \dfrac{r_i T_{\mathrm{win}} F_i}{q_i} \end{bmatrix}. \tag{5}$$

In Eq (5), $F_a$ is the Fano factor of the spike counts averaged over neurons in populations *a* = e, i over time windows of size $T_{\mathrm{win}}$. The second term in Eq (5) is $\mathcal{O}(1/N)$ and accounts for intrinsically generated covariability [23] within excitatory or inhibitory populations (note that this term does not refer to variances, but instead to mean covariances between spike trains in the same subpopulations). The matrix $\Gamma$ has the same structure as *C* and represents the covariance

between external inputs (See Baker et al., 2019 Appendix A for a detailed derivation of this term [23]).

If external neural activity is uncorrelated ($c_x = 0$), then

$$\Gamma = \overline{W}_x \overline{W}_x^T \frac{r_x}{q_x} \sim \mathcal{O}(1) \tag{6}$$

so that $C \sim \mathcal{O}(1/N)$, and the network is in an *asynchronous* regime. If external neural activity is correlated with mean pairwise correlation coefficient $c_x \neq 0$, then in leading order $N$,

$$\Gamma = N\overline{W}_x \overline{W}_x^T c_x r_x \sim \mathcal{O}(N), \tag{7}$$

so that $C \sim \mathcal{O}(1)$, and the network is in a *correlated* state. Eq (5) can be extended to cross–spectral densities as shown in S1 Appendix and by Baker et al. [23].

## Network model

For illustration, we used recurrent networks of $N$ exponential integrate–and–fire (EIF) neurons (See S1 Appendix), 80% of which were excitatory ($E$) and 20% inhibitory ($I$) [23, 24, 35, 54, 62]. The initial connectivity structure was random:

$$J_{jk}^{ab} = \frac{1}{\sqrt{N}} \begin{cases} j_{ab} & \text{with probability } p_{ab}, \\ 0 & \text{otherwise.} \end{cases} \tag{8}$$

Initial synaptic weights were therefore independent. We set $p_{ab} = 0.1$ for all $a$, $b$ = e, i, and denote by $J_{jk}^{ab}$ the weight of a synapse between presynaptic neuron $k$ in population $b$ = e, i, x and postsynaptic neuron $j$ in population $a$ = e, i. We modeled postsynaptic currents using an exponential kernel, $K_a(t) = \tau_a^{-1} e^{-t/\tau_a} H(t)$ for each $a$ = e, i, x where $H(t)$ is the Heaviside function.

**Synaptic plasticity rules.** To model activity–dependent changes in synaptic weights we used eligibility traces to define the propensity of a synapse to change [63–67]. The eligibility trace, $x_j^a(t)$, of neuron $j$ in population $a$ evolves according to

$$\tau_{\text{STDP}} \frac{dx_j^a(t)}{dt} = -x_j^a(t) + \tau_{\text{STDP}} S_j^a(t), \tag{9}$$

for $a$ = e, i, where $S_j^a(t) = \sum_n \delta(t - t_n^{a,j})$ is the sequence of spikes of neuron $j$. The eligibility trace, and the time constant, $\tau_{\text{STDP}}$, define a period following a spike in the pre– or post–synaptic cell during which a synapse can be modified by a spike in its counterpart.

Our theory of synaptic plasticity allows any synaptic weight to be subject to constant drift, changes due to pre– or post–synaptic activity only, and/or due to pairwise interactions in activity between the pre– and post–synaptic cells (zero, first, and second order terms, respectively, in Eq (10)). The theory can be extended to account for other types of interactions. Each synaptic weight therefore evolves according to a generalized STDP rule:

$$\frac{dJ_{jk}^{ab}}{dt} = \eta_{ab}\left(A_0 + \sum_{\alpha=\{a,j\},\{b,k\}} A_\alpha S_\alpha + \sum_{\alpha,\beta=\{a,j\},\{b,k\}} B_{\alpha,\beta} x_\alpha S_\beta\right) \tag{10}$$

where $\eta_{ab}$ is the learning rate that defines the timescale of synaptic weight changes, $A_0$, $A_\alpha$, $B_{\alpha\beta}$ are functions of the synaptic weight, $J_{jk}^{ab}$, and $a$, $b$ = e, i. For instance, the term $B_{(e,k),(i,j)} x_k^e S_j^i$ represents the contribution due to a spike in post–synaptic cell $j$ in the inhibitory subpopulation, at the value $x_k^e$ of the eligibility trace in the pre–synaptic cell $k$ in the excitatory subpopulation.

**Table 1. Examples of STDP rules.** A number of different plasticity rules can be obtained as special cases of the general form given in Eq (10).

| STDP Rule | Coefficients | Equation |
|---|---|---|
| Classical EE Hebbian [31, 32, 51] | $B_{(e,j),(e,k)} = -1$ <br> $B_{(e,k),(e,j)} = 1$ | $\frac{dJ^{ee}_{jk}}{dt} = \eta_{ee}\left(x^e_k S^e_j - x^e_j S^e_k\right)$ |
| Classical EE Anti-Hebbian [31, 32] | $B_{(e,j),(e,k)} = 1$ <br> $B_{(e,k),(e,j)} = -1$ | $\frac{dJ^{ee}_{jk}}{dt} = \eta_{ee}\left(-x^e_k S^e_j + x^e_j S^e_k\right)$ |
| Weight–dependent EE Hebbian [31, 32, 51] | $B_{(e,j),(e,k)} = -J^{ee}_{jk}$ <br> $B_{(e,k),(e,j)} = J_{\max}$ | $\frac{dJ^{ee}_{jk}}{dt} = \eta_{ee}\left(J_{\max} x^e_k S^e_j - J^{ee}_{jk} x^e_j S^e_k\right)$ |
| Homeostatic Inhibitory [41] | $A_{i,k} = \alpha_e \frac{J^{ei}_{jk}}{J_{norm}}$ <br> $B_{(e,j),(i,k)} = -\frac{J^{ei}_{jk}}{J_{norm}}$ <br> $B_{(i,k),(e,j)} = -\frac{J^{ei}_{jk}}{J_{norm}}$ | $\frac{dJ^{ei}_{jk}}{dt} = -\eta_{ei}\frac{J^{ei}_{jk}}{J_{norm}}\left[(x^e_j - \alpha_e)S^i_k + x^i_k S^e_j\right]$ |
| Oja's Rule [73] | $B_{(e,j),(e,j)} = -J^{ee}_{jk}$ <br> $B_{(e,j),(e,k)} = \beta$ | $\frac{dJ^{ee}_{jk}}{dt} = \eta_{ee}\left(\beta x^e_j S^e_k - J^{ee}_{jk} x^e_j S^e_j\right)$ |
| Kohonen's Rule [52] | $A_{e,j} = -J^{ee}_{jk}$ <br> $B_{(e,j),(e,k)} = \beta$ | $\frac{dJ^{ee}_{jk}}{dt} = \eta_{ee}\left(\beta x^e_k S^e_k - J^{ee}_{jk} S^e_j\right)$ |

Higher order interactions are at the heart of triplet rules [47, 68–70], and other types of interactions may also be important, e.g., for calcium–based update rules [71, 72]. For simplicity we here focus on pairwise interactions between spikes and eligibility traces, and leave extensions to more complex rules for future work.

This general formulation captures a range of classical plasticity rules as special examples: Table 1 shows that different choices of parameters yield Hebbian [31, 32, 51], anti–Hebbian, as well as Oja's [73], and other rules (See Fig A in S1 Appendix for illustrations of the STDP function of each rule in Table 1). The BCM rule [68], and other rules [69, 70] that depend on interactions beyond second order will be considered elsewhere.

### Dynamics of mean synaptic weights in balanced networks

To understand how the dynamics of the network, and synaptic weights co–evolve we derived effective equations for the firing rates, spike count covariances, and synaptic weights using Eqs (3) and (5). The following is an outline, and details can be found in S1 Appendix.

We assumed that changes in synaptic weights occur on longer timescales than the dynamics of the eligibility traces and the correlation timescale, i.e., $1/\eta_{ab} \gg T_{win}$ [30, 38–40, 45, 74]. Let $\Delta T$ be a time increment larger than the timescale of eligibility traces, $\tau_{STDP}$, and $T_{win}$, but smaller than $1/\eta_{ab}$, so that the difference quotient of the weights and time is given by [30]:

$$\frac{\Delta J^{ab}_{jk}}{\Delta T} = \frac{\eta_{ab}}{\Delta T}\int_0^{\Delta T}\left[A_0 + \sum_{\alpha=\{a,j\},\{b,k\}} A_\alpha S_\alpha + \sum_{\alpha,\beta=\{a,j\},\{b,k\}} B_{\alpha,\beta} x_\alpha S_\beta\right]dt. \tag{11}$$

The difference in timescales allows us to assume that the firing rates and covariances are in quasi–equilibrium. We used $1/\eta_{ab} = 10^5$ ms, and $\tau_{STDP} = 200$ ms, with correlation time window width $T_{win} = 250$ ms. Our derivations require $\tau_{STDP} \ll \Delta T \ll 1/\eta_{ab}$, however an exact numerical value for $\Delta T$ is neither used nor needed (See S1 Appendix: "What happens when timescales are not separated?"). Replacing the terms on the right hand side of Eq (11), with their averages over time, and over different network subpopulations, we obtain the following

mean–field equation for the weights:

$$\frac{dJ_{ab}}{dt} = \eta_{ab}\left(A_0 + \sum_{\alpha,\beta=\{a,b\}} \text{Rate}_{\alpha,\beta} + \text{Cov}_{\alpha,\beta}\right), \tag{12}$$

where

$$\text{Rate}_{\alpha,\beta} = A_\alpha r_\alpha/2 + B_{\alpha,\beta}\tau_{STDP}r_\alpha r_\beta, \tag{13}$$

$$\text{Cov}_{\alpha,\beta} = B_{\alpha,\beta}\int_{-\infty}^{\infty} \tilde{K}(f)\langle S_\alpha, S_\beta\rangle(f)df, \tag{14}$$

and $\tilde{K}(f)$ is the Fourier transform of the synaptic kernel, $K(t)$. Recall that $\langle S_\alpha, S_\beta\rangle(f)$ is the average cross spectral density of spike trains in populations $\alpha, \beta$. The cross spectral density (CSD) of a pair of spike trains is defined as the Fourier transform of the covariance function between the two spike trains, and when evaluated at $f = 0$, the CSD is proportional to the spike count covariance between the two spike trains (See S1 Appendix).

For example, classical Hebbian *EE* plasticity in Table 1 leads to the following mean–field equation,

$$\frac{dJ_{ee}}{dt} = \eta_{ee}(J_{\max} - J_{ee})\left(\tau_{STDP}r_e^2 + \int_{-\infty}^{\infty} \tilde{K}(f)\langle S_e, S_e\rangle(f)df\right). \tag{15}$$

Eqs (3), (5) and (12) thus self–consistently describe the macroscopic dynamics of the balanced network. There are two approaches to analyzing this coupled system of ordinary differential equations: (1) solve directly for the steady–states of the system; or (2) apply numerical integration to obtain the evolution of the system in time. To obtain the equilibria, we first find the firing rates and covariances (both in terms of plastic weight $J_{ab}$) obtained using the mean–field description of the balanced network, Eqs (3) and (5). We next substitute the results into Eq (12), set $\frac{dJ_{ab}}{dt} = 0$, and find the roots. We denote the solution by $J_{ab}^*$. We then use the mean synaptic weight (root of Eq (12), $J_{ab}^*$) to obtain the corresponding rates and covariances using Eqs (3) and (5). Alternatively, we can solve the system iteratively over time and obtain the time evolution of rates, covariances, and weights. Starting at an arbitrary value of $J_{ab}(t)$, we proceed in the same way as in the first approach, but instead of setting $\frac{dJ_{ab}}{dt} = 0$, we use $J_{ab}(t)$ to compute the value of the derivative at time $t$, $\frac{dJ_{ab}}{dt}|_t$, and use it to update the mean weight at the next time step, $J_{ab}(t + \Delta T)$. We then use $J_{ab}(t + \Delta T)$ to update rates and covariances. We repeat this process until convergence (See S1 Appendix: "Transient dynamics of synaptic weights" for sample trajectories under different rules, and for our criterion to determine stationarity of synaptic weights).

## Perturbative analysis

We next show how rates and spike count covariances are impacted by perturbations in synaptic weights. At steady state the average firing rates in a balanced network with mean–field connectivity matrix $\overline{W}_0$ are given by

$$\boldsymbol{r}_0 = -\overline{W}_0^{-1}\overline{W}_x r_x \tag{16}$$

We assume that the mean–field connectivity matrix is perturbed to $\overline{W}_{\text{perturb}} = \overline{W}_0 + \Delta\overline{W}$. Using Neumann's approximation [75], $(I + H)^{-1} \approx (I - H)$, which holds for any square matrix

$H$ with $\|H\| < 1$, and ignoring terms of power 2 and larger, we obtain,

$$\overline{W}^{-1}_{\text{perturb}} \quad = (\overline{W}_0 + \Delta\overline{W})^{-1} = (\overline{W}_0(I + \overline{W}_0^{-1}\Delta\overline{W}))^{-1} \tag{17}$$

$$\approx (I - \overline{W}_0^{-1}\Delta\overline{W})\overline{W}_0^{-1}, \tag{18}$$

where $I$ is the identity matrix of appropriate size. We use this approximation of the perturbed weights to estimate the rates and spike count covariances using Eqs (3) and (5). The $2 \times 2$ mean–field connectivity matrix, $\overline{W}_0$, must be non–singular for the balanced state to exist and for Neumann's approximation to hold [13]. While the non–singularity of $\overline{W}_0$ is a non–restrictive condition for two neural populations, $\overline{W}_0$ can become singular in some models with several neural sub–populations [49, 54].

## Comparison of theory with numerical experiments

We define spike trains of individual neurons in the population as sums of Dirac delta functions, $S_i(t) = \sum_j \delta(t - t_{ij})$, where the $t_{ij}$ is the time of the $j^{th}$ spike of neuron $i$. Assuming the system has reached equilibrium, we partition the interval over which activity has been measured into $K$ equal subintervals, and define the spike count covariance between two cells as,

$$\text{cov}(n_{1k}, n_{2k}) = \sum_k (n_1^k - \overline{n}_1)(n_2^k - \overline{n}_2), \tag{19}$$

where $n_{ik}$ is the spike count of neuron $i$ in subinterval, or time window, $k$, and $\overline{n}_i = \frac{1}{K}\sum_k n_{ik}$ is the average spike count over all subintervals. In simulations we used subintervals of size $T_{\text{win}} =$ 200 ms, although the theory applies to sufficiently long subintervals, and can be extended to shorter intervals as well. The spike count covariance thus captures shared fluctuations in firing rates between the two neurons [76].

## Results

We next apply the theory described in the Methods to show how synaptic weights coevolve with firing rates in balanced networks under different plasticity rules. We start with an example of excitatory plasticity which has been the main focus of experimental and theoretical studies, and show that our theory can be used to determine the stability of balanced networks under commonly used excitatory STDP rules. More recently, inhibitory plasticity has been proposed to play an important role in regulating the dynamics of neural networks. Our approach provides a theoretical foundation for some of these findings. Finally, we show that our theory can be used to make experimental predictions by considering a plastic network under optogenetic stimulation, and demonstrating that our framework can describe the dynamics of networks in such biologically relevant regimes.

## Balanced networks under excitatory plasticity

Excitatory plasticity plays a central role in theories of learning, but can lead to instabilities [31, 32, 34, 44]. Our theory predicts the stability of the balanced state, the fixed point of the system, and the effect the plasticity rule on the dynamics of the network.

We consider a network in a correlated state with excitatory–to–excitatory (*EE*) weights that evolve according to Kohonen's rule [52, 77]. This rule was first introduced in artificial neural networks [78], and was later shown to lead to the formation of self–organizing maps in model biological networks. [78, 79] We use our theory to show that Kohonen's rule leads to stable asynchronous or correlated balanced states, and verify these predictions in simulations.

Kohonen's Rule can be implemented by letting *EE* synaptic weights evolve according to [52] (See Table 1),

$$\frac{dJ_{jk}^{ee}}{dt} = \eta_{ee}\left(\beta x_j^e S_k^e - J_{jk}^{ee} S_j^e\right), \tag{20}$$

where $\beta > 0$ is a parameter that can change the fixed point of the system (See S1 Appendix: "Saddle–node bifurcation of excitatory weights in Kohonen's STDP rule"). This STDP rule is competitive as weight updates only occur when the pre–synaptic neuron is active, so that the most active pre–synaptic neurons change their synapses more than less active pre–synaptic cells.

The mean–field approximation describing the evolution of synaptic weights given in Eq (12) has the form:

$$\frac{dJ_{ee}}{dt} = \eta_{ee}(\beta \tau_{STDP} r_e^2 - J_{ee} r_e + \beta \int_{-\infty}^{\infty} \tilde{K}(f)\langle S_e, S_e \rangle df). \tag{21}$$

The fixed point of Eq (21) can be obtained by using the expressions for the rates and covariances obtained in the balanced state (Eqs (3) and (5)). The rates and covariances at steady–state can then be obtained from the resulting weights.

**Equilibria of correlated balanced networks under excitatory STDP.** Our theory predicts that the network attains a stable balanced state, and the average rates, weights, and covariances at this equilibrium (Fig 1) (See S1 Appendix: "Statistics and dynamics of balanced networks under pairwise STDP rules" for empirical distributions under Kohonen's and other rules). These predictions agree with numerical simulations in both the asynchronous and correlated states (Fig 1B and 1C). As expected, predictions improve with network size, *N*, and spike count covariances scale as $1/N$ in the asynchronous state (Fig 1D–1F). Similar agreement holds in the correlated state, including the impact of the correction introduced in Eq (21) (Fig 1G–1I).

The predictions of the theory hold in all cases we tested (See S1 Appendix: "Asymptotic behavior in weight–dependent Hebbian STDP"). Understanding when plasticity will support a stable balanced state allows one to implement Kohonen's rule in complex contexts and tasks, without the emergence of pathological states (See S1 Appendix: "Classical Hebbian STDP leads to unstable dynamics").

**Dynamics of correlated balanced networks under excitatory STDP.** We next asked whether and how the equilibrium and its stability are affected by correlated inputs to a plastic balanced network. In particular, we used our theory to determine whether changes in synaptic weights are driven predominantly by the firing rates of the pre– and post–synaptic cells, or correlations in their activity. We also asked whether correlations in neural activity can change the equilibrium, the speed of convergence to the equilibrium, or both?

We first address the role of correlations. As shown in the previous section, our theory predicts that a plastic balanced network remains stable under Kohonen's rule, and an increase in the mean *EE* weights by 10–20% when input correlations are increased. Both predictions were confirmed by simulations (Fig 2A and 2B). The theory also predicted that this increase in synaptic weights results in negligible changes in firing rates, which simulations again confirmed (Fig 2C).

How large is the impact of correlations in plastic balanced networks more generally? To address this question, we assumed that only pairwise interactions affect *EE* synapses, as first order interactions depend only on rates after averaging. We thus set $B_{\alpha,\beta} \equiv 1$, and all other coefficients to zero in Eq (10). While the network does not reach a stable state under this arbitrary plasticity rule, it allows us to estimate the contribution of rates and covariances to the

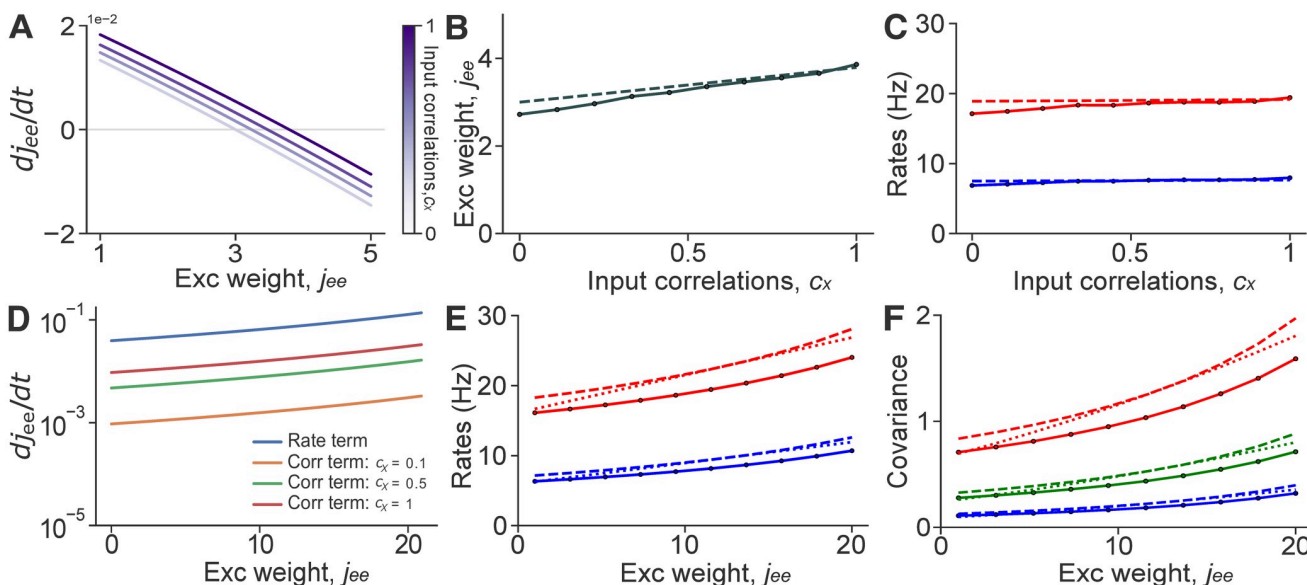

**Fig 2. Spike count covariances mildly impact the fixed point of synaptic weights and firing rates.** **A**: The rate of change of *EE* weights as function of the weight, $j_{ee}$, at different levels of input correlations, $c_x$. **B**: Mean steady–state *EE* synaptic weight for a range of input correlations, $c_x$. **C**: Mean *E* and *I* firing rates as a function of input correlations. **D**: Same as (**A**) but for an *EE* STDP rule with all coefficients involving order 2 interactions set equal to 1, and all other coefficients set equal to zero. **E**: Mean *E* and *I* firing rates as a function of mean *EE* synaptic weights. **F**: Mean spike count covariances between *E* spike trains, *I* spike trains, and between *E–I* spike trains as a function of *EE* synaptic weight, $j_{ee}$. Solid lines represent simulations (except in **A**, **D**), dashed lines are values obtained from theory (Eqs (3), (5) and (21)), and dotted lines were obtained from the perturbative analysis. Note that in all panels, 'Exc weight' refers to $j_{ee}$ rather than $J_{ee}$, as the former does not depend on $N$.

evolution of synaptic weights. Here $B_{\alpha, \beta}$ can have any nonzero value, since it scales both the rate and covariance terms. Under these conditions, our theory predicts that the rate term is at least an order of magnitude larger than the correlation term (even when rates themselves are small, *i.e.*, when $j_{ee}$ is small), and so correlations only have a low impact on the dynamics of synaptic weights (Fig 2D). Therefore, our theory predicts that, in general, changes in synaptic weights will largely be driven by changes in firing rate patterns, rather than changes in pairwise correlations.

We next ask the opposite question: How do changes in synaptic weights impact firing rates, and covariances? The full theory (see Eqs (3) and (5), and perturbative analysis in Materials and Methods) predict that the potentiation of *EE* weights leads to large increases in rates and spike count covariances. This prediction was again confirmed by numerical simulations (Fig 2E and 2F). This observation holds generally, and STDP rules that result in large changes in synaptic weights will produce large changes in rates and covariances.

Our theory thus shows that *in general* weight dynamics can be moderately affected by correlations when these are large enough (See S1 Appendix: "General impact of correlations in weight dynamics" for a similar analysis on Classical Hebbian STDP). In turn, changes in synaptic weights will generally change the structure of correlated activity in a balanced network.

## Balanced networks under inhibitory plasticity

Next, we show that in its basic form our theory can fail in networks subject to inhibitory STDP, and how the theory can be extended to capture such dynamics. The failure is due to correlations between weights and pre–synaptic rates which are typically ignored [13, 14, 23, 48–50], but can cause the mean–field description of network dynamics to become inaccurate.

This is similar to the breakdown of balanced state theory in the presence of correlations between in– and out–degrees discussed by Vegué and Roxin, 2019 [80].

To illustrate this, we consider a balanced network subject to homeostatic plasticity [41]. This type of plasticity has been shown to stabilize the asynchronous balanced state and conjectured to play a role in the maintenance of memories [35, 41, 81]. Following [41] we assume that *EI* weights evolve according to:

$$\frac{dJ_{jk}^{ei}}{dt} = -\eta_{ei} \frac{J_{jk}^{ei}}{J_{norm}} \left[ (x_j^e - \alpha_e)S_k^i + x_k^i S_j^e \right] \tag{22}$$

where $\alpha_e$ is a constant that determines the target firing rates of *E* cells and $J_{norm} \sim \mathcal{O}(1/\sqrt{N})$ is a normalization constant. Note that $J_{norm} < 0$ so the fraction in Eq (22) is positive assuming $J_{jk}^{ei} < 0$. In a departure from the rule originally proposed by Vogels et al. [41], we chose to multiply the time derivative by the current weight value. This modification creates an unstable fixed point at zero, prevents *EI* weights from changing signs, and keeps the analysis mathematically tractable (See S1 Appendix: "Modification to the inhibitory STDP rule" for details). An alternative way to prevent weights from changing sign would be to place a hard bound at zero, but this would create a discontinuity in the vector field of $J_{ei}$, complicating the analysis.

Under the rule described by Eq (22) a lone pre–synaptic spike depresses the synapse, while near–coincident pre– and post–synaptic spikes potentiate the synapse (See Fig A in S1 Appendix). Changes in *EI* weights steer the rates of individual excitatory cells to the target value $\rho_e := \frac{\alpha_e}{2\tau_{STDP}}$. Indeed, individual *EI* weights are potentiated if post–synaptic firing rates are higher than $\rho_e$, and depressed if the rate is below $\rho_e$. Our theory predicts that the network converges to a stable balanced state (Fig 3A). Correlations again have only a mild impact on the evolution of synaptic weights (Fig 3A).

Although our theory predicts a single stable fixed point for the average *EI* weight, simulations show that weights converge to a different average depending on the initial *EI* weights (Fig 3B–3E solid lines). A manifold of stable fixed points emerges due to synaptic competition, which is a consequence of heterogeneity in inhibitory firing rates in the network: Weights of highly active pre–synaptic inhibitory cells are potentiated more strongly compared to those of lower firing cells (Fig 3E). Thus while inhibitory rates and *EI* weights are initially uncorrelated, correlations emerge as the excitatory rates approach their target. Networks with different initial *EI* synaptic weights, converge to different final distributions, and the emergent correlations between weights and rates drive the system to different fixed points (Fig 3C and 3D).

We used a semi–analytical approach to confirm that correlations between weights and rates explain the discrepancy between predictions of the mean field theory, and simulations. To do so we introduced a correlation dependent correction term into the expression for the rates:

$$\lim_{N \to \infty} \vec{r} = -\overline{W}^{-1}(\overline{W}_x r_x + \text{cov}(J_{ei}, r_i)), \tag{23}$$

where $\text{cov}(J_{ei}, r_i) := [\langle \langle J_{jk}^{ei} r_k^i \rangle_k - \langle J_{jk}^{ei} \rangle_k \langle r_k^i \rangle_k \rangle_j, 0]^T$. The average covariances between weights and rates obtained numerically explain the departure from the mean–field predictions (Fig 3C). Using the corrected equation (Eq (23)) predicts mean equilibrium weights that agree well with simulations (Fig 3C dashed line).

We next asked whether the mean–field theory provides a good description of network dynamics in the absence of correlations between weights and rates. Such correlations disappear in a network with homogeneous inhibitory firing rates. Finding an initial distribution of weights that result in a balanced state with uniform inhibitory firing rates is non–trivial, and may not be possible outside of unstable regimes exhibiting rate–chaos where mean–field

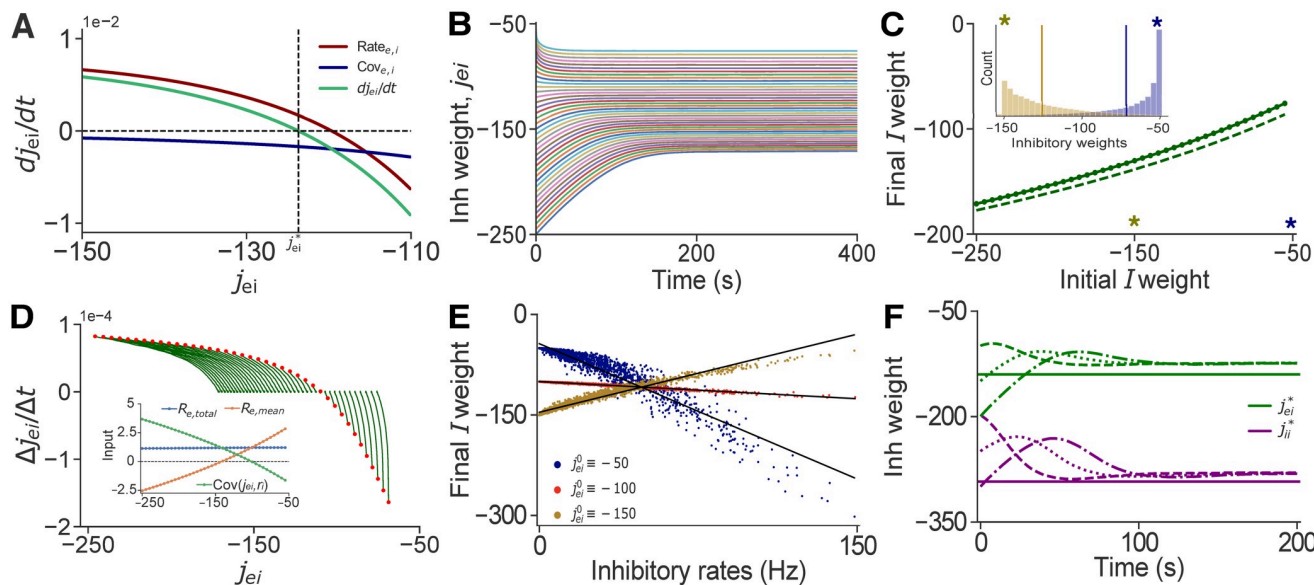

**Fig 3. Correlations between synaptic weights and inhibitory rates lead to the formation of a manifold in weight space. A**: The rate of change of *EI* weights as a function of the weights themselves. The contributions of the covariance (blue) is considerably smaller than the contribution of the rate (red), and the theory predicts a stable fixed point. **B**: Evolution of inhibitory weights showing that different initial weights converge to different fixed points. Also, weights starting at different initial conditions converge to equilibrium at different times for fixed $1/\eta_{ei} = 10^5$ ms. **C**: A manifold of fixed points in $j_{ei}^* - j_{ei}^0$ space emerges due to correlations between weights and rates. Solid line represents simulations, dashed line are values obtained from the modified theory (Eqs (23) and (5), and mean–field equation for weights under iSTDP in Table 1). Inset: Final distribution of *EI* weights for a network with initial weights $j_{ei}^0 = -150$ (yellow), and $j_{ei}^0 = -50$ (blue). Modified theory predicts the manifold of fixed points. **D**: Same as **A**, but obtained from simulations. Lines represent trajectories from different initial weights (red dots). Inset: Total recurrent input to *E* neurons, $R_{e,total} = \langle\langle J_{jk}^{ee} r_j^e\rangle_j + \langle J_{kl}^{ei} r_l^i\rangle_l\rangle_k$ for a range of initial weights. Mean recurrent input to *E* cells, $R_{e,mean} = \overline{w}_{ee} r_e + \overline{w}_{ei} r_i$. The mean input deviates from the total input due to emergent correlations between weights and rates, $Cov(J_{ei}, r_i) = R_{e,total} - R_{e,mean}$. **E**: The weights of individual *EI* synapses corresponding to the same post–synaptic *E* cell as a function of the equilibrium firing rates of pre–synaptic *I* neurons. Each color represents a different simulation of the network with different initial *EI* weight. Equilibrium inhibitory weights and pre–synaptic rates are correlated (Blue: $R^2 = 0.952$, Red: $R^2 = 0.9865$, Yellow: $R^2 = 0.979$). **F**: Sample trajectories of the $j_{ei} - j_{ii}$ system for a network of $N = 10^4$ neurons in an asynchronous state. Simulations with different initial weights (dashed lines), converge to a fixed point close to the one predicted by the theory (solid line).

theory ceases to be valid [82]. However, allowing *II* synapses to evolve under the same plasticity rule we used for *EI* synpases can homogenize inhibitory firing rates: If we let

$$\frac{dJ_{jk}^{ii}}{dt} = -\eta_{ii} \frac{J_{jk}^{ii}}{J_{norm}} \left[ (x_j^i - \alpha_i)S_k^i + x_k^i S_j^i \right],$$ (24)

all inhibitory responses approach a target rate $\rho_i = \frac{\alpha_i}{2\tau_{STDP}}$, effectively removing the variability in *I* rates. The evolution of the mean *II* and *EI* synaptic weights is now given by

$$\begin{aligned}
\frac{dJ_{ei}}{dt} &= -\eta_{ei} \frac{J_{ei}}{J_{norm}} \left( (2\tau_{STDP} r_e - \alpha_e) r_i + 2 \int_{-\infty}^{\infty} \tilde{K}(f) \text{Re}[\langle S_e, S_i \rangle] df \right), \\
\frac{dJ_{ii}}{dt} &= -\eta_{ii} \frac{J_{ii}}{J_{norm}} \left( (2\tau_{STDP} r_i - \alpha_i) r_i + 2 \int_{-\infty}^{\infty} \tilde{K}(f) \langle S_i, S_i \rangle df \right).
\end{aligned}$$ (25)

We conjectured that if inhibitory rates converge to a common target, synaptic competition would be eliminated, and no correlations between weights and rates would emerge. This in turn would remove the main obstacle to the validity of a mean–field description. The fixed point of these equations can again be obtained using Eqs (3) and (5) which predict that the

network remains in a stable balanced state (asynchronous or correlated). We also require $\eta_{ei} \geq \eta_{ii}$, since when $\eta_{ei}$ is much slower than $\eta_{ii}$, the network becomes unstable as homogeneous inhibitory weights and rates are not able to stabilize the heterogeneous distribution of $E$ activity (See S1 Appendix: "Stability of iSTDP in $EI$ and $II$ connections."). We chose the same STDP timescale for both $EI$ and $II$ synapses, and our predictions agree with the results of simulations (Fig 3F). The stable manifold of fixed points is replaced by a single stable fixed point, and the average weights and rates approach a state that is independent of the initial weight distribution.

This model of inhibitory plasticity is likely a large oversimplification. Synapses of different interneuron subtypes are likely subject to different plasticity rules operating on different timescales [17, 83], and would therefore not lead to uniform inhibitory firing rates. The mean–field theory we presented here can be extended to account for multiple inhibitory subtypes with different plasticity rules.

We next show that the balanced network subject only to $EI$ plasticity is robust to perturbatory inputs. Our theory predicts, and simulations confirm, that this learning rule maintains balance when non–plastic networks do not, and it can return the network to its original state after stimulation.

## Inhibitory plasticity adapts response to stimuli

Thus far, we analyzed the dynamics of plastic networks in isolation. However, cortical networks are constantly driven by sensory input, as well as feedback from other cortical and sub–cortical areas. We next ask whether and how balance is restored if a subset of pyramidal neurons are stimulated [54].

In experiments using optogenetics not all target neurons express the channelrhodopsin 2 (ChR2) protein [84–87]. Thus stimulation separates the target, e.g., pyramidal cell population into stimulated and unstimulated subpopulations. Although classical mean–field theory produced singular solutions, Ebsch et al. showed that the theory can be extended, and that a non–classical balanced state is realized: Balance at the level of population averages ($E$ and $I$) is maintained, while balance at the level of the three subpopulations is broken [54]. Since local connectivity is not tuned to account for the extra stimulation (optogenetics), local synaptic input cannot cancel external input to the individual subpopulations. However, the input averaged over the stimulated and unstimulated excitatory population is cancelled.

We show that inhibitory STDP, as described by Eq (22), can restore balance in the inputs to the stimulated and unstimulated subpopulations. Similarly, Vogels et al. showed numerically that such plasticity restores balance in memory networks [41]. Here, we present an accompanying theory that describes the evolution of rates, covariances, and weights before, during, and after stimulation, and confirm the prediction of the theory numerically.

We assume that a subpopulation of pyramidal neurons in a correlated balanced network receives a transient excitatory input. This could be a longer transient input from another sub-network, or an experimentally applied stimulus. To model this drive, we assume that the network receives input from two populations of Poisson neurons, $X_1$ and $X_2$. The first population drives all neurons in the recurrent network, and was denoted by $X$ above. The second population, $X_2$, provides an additional input to a subset of excitatory cells in the network, for instance ChR2 expressing pyramidal neurons ($E_{expr}$ in Fig 4). The resulting connectivity matrix between the stimulated ($e_1$), unstimulated ($e_2$) and inhibitory (i) subpopulations, and the feed–

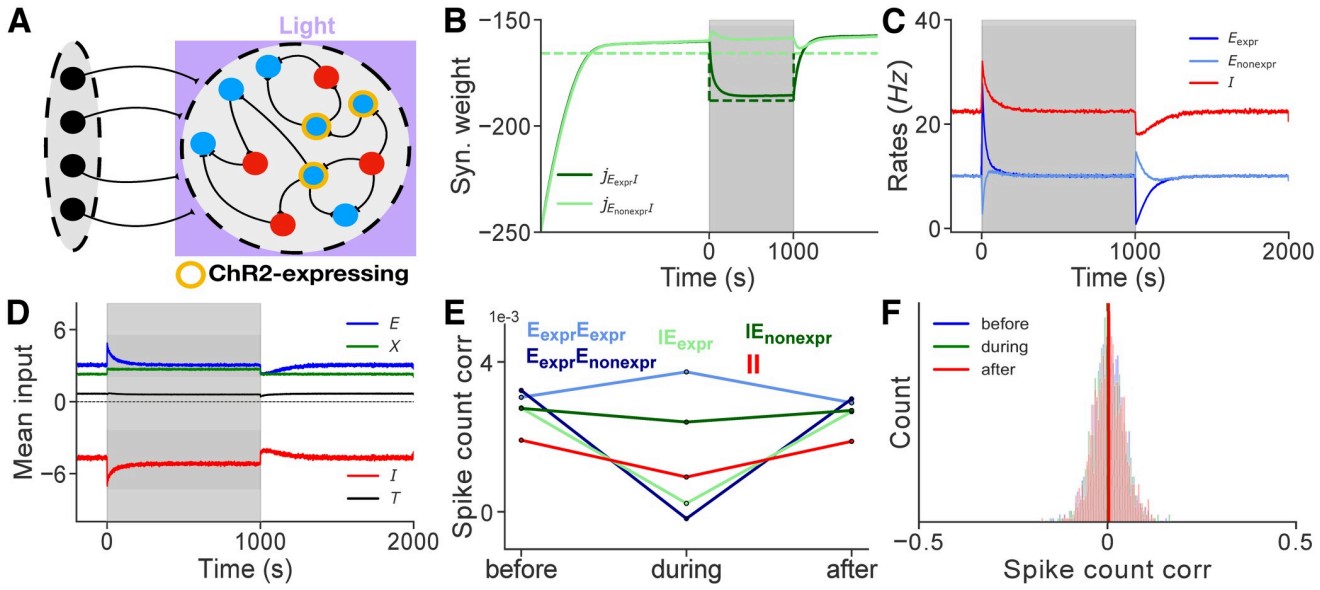

**Fig 4. Framework for STDP in balanced networks describes the dynamics of networks receiving optogenetic input. A**: A recurrent network of excitatory, $E$, and inhibitory, $I$, neurons is driven by an external feedforward layer $X_1$ of uncorrelated Poisson neurons. Neurons that express ChR2 are driven by optogenetic input, which is modeled as an extra layer of Poisson neurons denoted by $X_2$. **B**: Evolution of mean synaptic weights over the course of the experiment. **C**: Evolution of mean firing rates. Inhibitory STDP maintains $E$ rates near the target, $\frac{\alpha_e}{2\tau_{STDP}}$. **D**: Evolution of mean excitatory, external, inhibitory, and total currents. Balance is transiently disrupted at stimulus onset and offset, but it is quickly restored by iSTDP. **E**: Mean spike count correlations before, during, and after stimulation remain very weak for all pairs. **F**: The distribution of spike count correlations also remains nearly unchanged with weak mean correlations before, during, and after stimulation. Solid lines represent simulations, dashed lines are values obtained from theory (Eqs (3), (5), (27) and (28)).

forward input weight matrix have the form:

$$\overline{W} = \begin{bmatrix} \overline{w}_{e_1 e_1} & \overline{w}_{e_1 e_2} & \overline{w}_{e_1 i} \\ \overline{w}_{e_2 e_1} & \overline{w}_{e_2 e_2} & \overline{w}_{e_2 i} \\ \overline{w}_{ie_1} & \overline{w}_{ie_2} & \overline{w}_{ii} \end{bmatrix}, \text{ and } \overline{W}_x = \begin{bmatrix} \overline{w}_{e_1 x_1} & \overline{w}_{e_1 x_2} \\ \overline{w}_{e_2 x_1} & 0 \\ \overline{w}_{ix_1} & 0 \end{bmatrix}, \tag{26}$$

where $\overline{w}_{ab} = p_{ab} j_{ab} q_b \sim \mathcal{O}(1)$, as before.

The mean–field equation relating firing rates to average weights and input (Eq (3)) holds, with the vector of rates $r = [r_{e_1}, r_{e_2}, r_i]^T$, and input vector $r_x = [r_{x_1}, r_{x_2}]^T$. Similarly, mean spike count covariances are now represented by a $3 \times 3$ matrix that satisfies Eq (5). The mean $E_1 I$ and $E_2 I$ weights evolved according to

$$\frac{dJ_{e_1 i}}{dt} = -\eta_{e_1 i} \frac{J_{e_1 i}}{J_{e_1 i}^{norm}} \left( (2\tau_{STDP} r_{e_1} - \alpha_e) r_i + 2 \int_{-\infty}^{\infty} \tilde{K}(f) \text{Re}[\langle S_{e_1}, S_i \rangle] df \right) \tag{27}$$

$$\frac{dJ_{e_2 i}}{dt} = -\eta_{e_2 i} \frac{J_{e_2 i}}{J_{e_2 i}^{norm}} \left( (2\tau_{STDP} r_{e_2} - \alpha_e) r_i + 2 \int_{-\infty}^{\infty} \tilde{K}(f) \text{Re}[\langle S_{e_2}, S_i \rangle] df \right). \tag{28}$$

We simulated a network of $N = 10^4$ neurons in an asynchronous state with $c_{x_1} = c_{x_2} = 0$. A subpopulation of 4000 $E$ cells receives transient input. Solving Eqs (27) and (28) predicts that inhibitory plasticity will alter $EI$ synaptic weights so that the firing rates of both the $E_{expr}$ and the $E_{non-expr}$ approach the target firing rate before, during, and after stimulation. Once the

network reaches steady state the mean inputs to each subpopulation cancel. Thus changes in *EI* weights restore balance at the level of individual subpopulations or "detailed balance," consistent with previous studies [41, 81]. Simulations confirm these predictions (Fig 4B–4D).

When the input is removed, the inhibitory weights onto cells in the $E_{\text{expr}}$ subpopulation converge to their pre–stimulus values, returning $E_{\text{expr}}$ rates to the target value, and reestablishing balance (Fig 4B–4D). Correlations remain low ($\mathcal{O} \sim 10^{-4}$) before, during, and after stimulation (Fig 4E and 4F), suggesting that at equilibrium the network is in the asynchronous state.

Our theory thus describes how homeostatic inhibitory STDP increases the stability and robustness of balanced networks to perturbations by balancing inputs at a level of individual cells, maintaining balance in regimes in which non–plastic networks cannot maintain balance. We presented an example in which only one subpopulation is stimulated. However, the theory can be extended to any number of subpopulations in asynchronous or correlated balanced networks receiving a variety of transient stimulus.

## Discussion

We have developed an analytical framework that predicts the impact of a general class of STDP rules on the weights and dynamics of balanced networks. The balanced state is generally maintained under synaptic weight changes, as long as the rates remain bounded. Additionally, we found that correlations in spiking activity can introduce a small shift in the steady state, and change how quickly the fixed point is reached.

One of the most important issues in understanding neural dynamics is establishing conditions under which the network remains active, yet stable as synaptic weights change. The theory we developed can help us address these questions, but it does have limitations. Since we used a mean–field approach, we can only capture first moments. While mean weight stability may not imply stable network dynamics (consider the case when weight variance diverges in Classical Hebbian STDP in S1 Appendix), instability in the mean weights does imply that the network is also unstable.

As we mentioned, our theory can be used to show that small modifications to weight updates can stabilize different STDP rules. The question remains whether Hebbian EE plasticity can be stabilized through an interaction with STDP rules at different synapses? For instance, Litwin–Kumar and Doiron used a triplet voltage STDP rule that was stabilized by hard constraints and weight normalization to produce network assemblies [35]. This rule by itself lead to stable but pathological behavior, and they introduced iSTDP to restore a balanced, asynchronous network state. While such voltage–based triplet rules are outside the scope of the present study, we could use extensions of the mean–field theory to describe the impact of second and higher order moments on the evolution of weights, and network dynamics [88]. Our theory suggests that the classical pairwise Hebbian STDP cannot be stabilized by other STDP rules such as iSTDP.

In the tight balance regime, large excitatory and inhibitory inputs cancel on average [15], resulting in a fluctuation–driven state exhibiting irregular spiking. This cancellation is achieved when synaptic weights are scaled by $1/\sqrt{N}$ and external input is strong [13, 14, 19, 89]. Our main assumption was that synaptic weights change slowly compared to firing rates. As this assumption holds generally, we believe that our approach can be extended to other dynamical regimes. For instance supralinear stabilized networks (SSNs) operate in a loosely balanced regime where the net input is comparable in size to the excitatory and inhibitory inputs, and firing rates depend nonlinearly on inputs. Balanced networks and SSNs can behave differently, as they operate in different regimes. However, as shown in [56], SSNs and balanced networks may be derived from the same model under appropriate choices of parameters. In

other words, the tight balanced solution can be realized in an SSN, and SSN–like solutions can be attained in a balanced network. This suggests that an extension of our theory of plasticity rules to SSNs should be possible.

We obtained a mean–field description of the balanced network by averaging over the entire inhibitory and excitatory subpopulation, and a single external population providing correlated inputs. As shown in the last section, the theory can naturally be extended to describe networks consisting of multiple physiologically or functionally distinct subpopulations, as well as multiple input sources.

The mean–field description cannot capture the effect of some second order STDP rules as synaptic competition can correlate synaptic weights and pre–synaptic rates. We have shown that this can lead to different initial weight distributions converging to different equilibria. This can be interpreted as the maintenance of a previous state of the network over time.

The present theory relies on a separation of timescales between spiking dynamics and weight changes. Such timescale separation is supported by a number of experiments [30–32, 90, 91]. We show in the Appendix (see S1 Appendix: "What happens when timescales are not separated?') that reducing this timescale separation, and increasing weight updates leads to a breakdown of the theory, and can result in network instability.

In mammalian brains, timescales of weight changes may not always be separated from rate and correlation timescales. The size and timescale of weight updates is likely to depend on many factors that can modulate STDP, such as spiking patterns, synapses type, brain area, network state, neuromodulation, and others. Separation of timescales may not be pronounced in certain non–cortical areas, such as the hippocampus, which can be rapidly modified [91]. For example, Petersen et al., 1998 and Froemke et al., 2006 found significant changes in putative synaptic weights over short timescales in hippocampal CA1/CA3 slices [92] and in visual cortical slices subject to multispike pre–and post–synaptic bursts [93], respectively. However, it is possible that the rate of change of synaptic weights may be overestimated *in vitro* [91].

How is our separation of timescales assumption affected when rapid compensatory processes are needed for homeostasis, given that experiments show that homeostasis is a process that is even slower than the timescale of STDP? Experimental evidence suggests that homeostatic processes can take hours or days [42, 81, 90, 91, 94–98]. On the other hand, theoretical models show that synaptic plasticity can be unstable in the presence of such slow homeostasis, and needs to be coupled with rapid compensatory processes such as inhibitory STDP [91, 94]. The separation of timescales in our theory still puts synaptic dynamics on the "fast" side of the spectrum, as it separates network dynamics that occur over milliseconds from weight dynamics that take place over seconds or minutes. Hence, the assumption of timescale separation is still valid in our implementation of homeostatic inhibitory plasticity.

In plastic networks, correlations between weights and other features such as in–degrees, or out–degrees can emerge [80]. We have shown how the theory can capture the case in which synaptic weights and pre–synaptic rates are correlated. While we were not able to find analytical expressions for these correlations, we showed that a second–order correction is sufficient to explain the observed dynamics. Eventually, the mean–field theory would need to be extended to account for higher order network motifs and their potential correlations with synaptic weights and firing rates. This might be possible by extending our approach, but we leave these extensions for future work.

We have assumed that connection probabilities are homogeneous which translates to a narrow distribution of in–degrees. Cortical networks are heterogeneous, and a broad distributions of in–degrees can break the classical balanced state [49, 50]. Balance can be restored with the introduction of homeostatic plasticity [49], or by including heterogeneous out–degrees correlated with in–degrees [50]. As we mentioned previously, in such cases our theory would need

to be extended to account for possible emerging correlations between weights and in-degrees or out–degrees. We relegate such extensions to future work.

A natural question that arises is why do correlations between weights and pre–synaptic rates only seem to play a role in iSTDP? In the examples of excitatory STDP we analyzed (Kohonen's rule and weight–dependent Hebbian rule), weights at equilibrium are determined by other parameters (Weight–dependent Hebbian rule) or rates (Kohonen's rule). Therefore weights are updated until those steady state values are achieved, yielding values independent of initial conditions. On the other hand, in the case of the inhibitory plasticity rule, inhibitory weights at equilibrium are determined by the firing rates alone. Since the firing rate vectors are lower–dimensional than the weight matrices, the equilibrium solution does not fully determine the weight matrices. This is shown in Fig 3C inset, where different distributions of weights can result in the same equilibrium firing rate when weights and pre–synaptic rates become correlated.

We have shown that different plasticity rules can result in distinct firing rate distributions in different subpopulations. As shown by Mongillo et al. this can result in an increase or decrease in sensitivity of activity patterns and memories to perturbation of different synapse classes [99].

Partial stimulation of a population of $E$ neurons has been shown to break balance due to the inability of the network in cancelling inputs when weights are static [54]. Ebsch et al. showed how classical balanced network theory can be modified to account for effects of input perturbations that break the classical balanced state [54]. Vogels et al. [41] (in addition to subsequent studies [35, 42, 81, 100–102]) showed empirically using simulations that inhibitory iSTDP can restore balance. We here provide a theoretical framework that describes the evolution of rates and weights before, during, and after a perturbation that breaks balance.

A number of mathematical theories have been proposed to describe the coevolution of weights, rates, and the structure of correlations under STDP in recurrent neural networks [37–39, 44–47, 74]. All of these approaches require knowledge of neurons' transfer functions (f-I curves and/or correlation susceptibility functions). Often neurons are assumed to be Poissonian, and their responses to inputs (f-I curves) are prescribed [37–39, 45–47, 74]. Other work [44] uses Fokker–Planck techniques to compute transfer functions. These approaches rely on an assumption that the input to each neuron is relatively weak or dominated by Gaussian white noise [103, 104]. Efficient, direct Fokker–Planck approaches are not available for two–dimensional integrate–and–fire models such as those with adaptation currents, though one–dimensional approximations have been derived [105, 106]. Some previous work [44] also assumes that STDP curves are approximately anti–symmetric, *i.e.*, there is a cancellation between the positive and negative parts of the curves (as in Panel A in Fig A in S1 Appendix).

Our approach uses balanced network theory to avoid the computation of transfer functions. As such, the resulting theory does not require an assumption of weak synaptic interactions or dominant Gaussian white noise input, but can be applied to networks with highly non–Gaussian, temporally correlated input (such as the networks in the correlated state considered here). Moreover, the balanced network theory we used is accurate for a range of neuron models, including those with adaptation currents [54, 107], and different STDP curves (as in Panels B–F in Fig A in S1 Appendix). However, balanced network theory relies on large $N$ asymptotics, which yielded accurate approximations for $N \sim 10,000$ in our case (Fig 1), but become less accurate in smaller networks. Our approach is not appropriate for modeling neural circuits that do not exhibit excitatory–inhibitory balance, such as observed in some disease states, some developmental stages, and in some sub–cortical neuronal networks. Finally, we used a mean–field approach that only yields approximations to population–averaged firing rates,

synaptic weights, and covariances, while other approaches [37–39, 44–47, 74] give approximations to these quantities at the level of individual neurons. Despite these limitations, our analytical approach was sufficient for answering the questions related to the interaction between excitatory–inhibitory balance, correlated neuronal activity, and plasticity that we considered.

We found that even in the correlated state, when the network receives temporally correlated input, changes in synaptic weights are dominated by firing rates, with correlations playing a secondary role (See Fig 2A and 2B). These findings are in agreement with previous work on STDP already mentioned before [44, 53]. Results by Ocker et al. were obtained in recurrent neural networks in different dynamical regimes and under different assumptions (See above for more details), while Graupner et al. used networks of two neurons with varying natural firing patterns.

The theoretical framework we presented is flexible, and can describe more intricate dynamics in circuits containing multiple inhibitory subtypes, and multiple plasticity rules, as well as networks in different dynamical regimes. Moreover, the theory can be extended to plasticity rules that depend on third order interactions [69, 70], such as the BCM rule [68]. This may produce richer dynamics, and change the impact of correlations.

## Conclusion

We developed a second order theory of spike–timing dependent plasticity for classical asynchronous, and correlated balanced networks [13, 14, 19, 23]. Assuming that synaptic weights change slowly, we derived a set of equations describing the evolution of firing rates, correlations as well as synaptic weights in the network. We showed that, when the mean–field assumptions are satisfied, these equations accurately describe the network's state, stability, and dynamics. However, some plasticity rules, such as inhibitory STDP, can introduce correlations between synaptic weights and rates. Although these correlations violate the assumptions of mean–field theory, we showed how to account for, and explain their effects. Additional plasticity rules can decorrelate synaptic weights and rates, reestablishing the validity of classical mean–field theory. Lastly, we showed that inhibitory STDP allows networks to maintain balance, and preserves the network's structure and dynamics when subsets of neurons are transiently stimulated. Our approach is flexible and can be extended to capture interactions between multiple populations subject to different plasticity rules.

## Supporting information

**S1 Appendix. Review of mean–field theory in balanced networks and supporting results.** This supplementary text contains (1) a review of classical mean–field theory of firing rates and spike count covariances in balanced networks; (2) the derivation of the equation that describes mean synaptic weights, a derivation of conditions under which synaptic weights do not change signs when undergoing inhibitory STDP, and general remarks on how synaptic weights can be affected by changes in rates or covariances; and (3) supporting results on separation of timescales, synaptic weight transient dynamics, stability of weights under Kohonen's rule, statistics and stability of synaptic weights under several STDP rules, the general impact of correlations in synaptic weights, a network undergoing iSTDP where synaptic weights change signs, and stability of iSTDP on *EI* and *II* synaptic weights. **Fig A**. **STDP windows of different plasticity rules**. **a**: Change in synaptic weights as a function of the relative timing of pre– and post–synaptic spikes in Classical Hebbian STDP (same as weight–dependent Hebbian). **b**: Same as **a**, but for inhibitory STDP. **c**: Same as **a**, but for Kohonen's rule when weights are below

parameter $\beta$. **d**: Same as **c**, but for the case when weights are above $\beta$. **e**: Same as **a**, but for Oja's rule when weights are below parameter $\beta$. **f**: Same as **e**, but for the case when weights are above $\beta$.
(PDF)

## Author Contributions

**Conceptualization:** Alan Eric Akil, Robert Rosenbaum, Krešimir Josić.

**Data curation:** Alan Eric Akil.

**Formal analysis:** Alan Eric Akil, Robert Rosenbaum, Krešimir Josić.

**Funding acquisition:** Robert Rosenbaum, Krešimir Josić.

**Investigation:** Krešimir Josić.

**Methodology:** Alan Eric Akil.

**Project administration:** Krešimir Josić.

**Software:** Alan Eric Akil, Robert Rosenbaum.

**Supervision:** Robert Rosenbaum, Krešimir Josić.

**Validation:** Alan Eric Akil, Krešimir Josić.

**Visualization:** Alan Eric Akil.

**Writing – original draft:** Alan Eric Akil, Krešimir Josić.

**Writing – review & editing:** Alan Eric Akil, Robert Rosenbaum, Krešimir Josić.

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
