## [Decision Letter · Decision Letter 0]

27 Jul 2020

Dear Dr. Josić,

Thank you very much for submitting your manuscript "Synaptic Plasticity in Correlated Balanced Networks" for consideration at PLOS Computational Biology.

As with all papers reviewed by the journal, your manuscript was reviewed by members of the editorial board and by several independent reviewers. In light of the reviews (below this email), we would like to invite the resubmission of a significantly-revised version that takes into account the reviewers' comments. In particular, we would like to see how you address the comment regarding the novelty of this work, and how the work fits in the context of other related work in the field, as noted by Rev 2. As an example, the Introduction currently does not place the work in the context of other published work, and 2/3 of the introduction is basically an extended abstract. 

We cannot make any decision about publication until we have seen the revised manuscript and your response to the reviewers' comments. Your revised manuscript is also likely to be sent to reviewers for further evaluation.

Sincerely,

Julijana Gjorgjieva, PhD

Guest Editor

PLOS Computational Biology

Kim Blackwell

Deputy Editor

PLOS Computational Biology

Reviewer's Responses to Questions

**Comments to the Authors:**

Reviewer #1: Akil et al. develop a mean field theory (with some important semi analytical corrections) for the co-evolution of synaptic weights and first and second order firing rate statistics in spiking neuronal networks. This is potentially an important contribution to both theoretical and experimental neuroscience, where the understanding of the effects various plasticity rules have at the network level are lacking.

This paper puts our understanding of these effects on more firm analytical grounds.

However, I think that the results could be strengthened significantly if the authors leverage the theory they developed to address more fundamental problems than the applications included in the current version.

Beyond that, the current version of the manuscript is missing some key references to and discussion of both theoretical and experimental literature that would help readers understand the importance of the results and their limitations.

major concern

A fundamental problem in theoretical neuroscience is the stability of neuronal network activity subject to activity dependent plasticity. It is clear--and the authors mention it in their paper (l. 217)-- that Hebbian excitatory plasticity leads to instabilities. The theory that the authors develop is exactly the analytical tool needed to understand what are minimal and/or realistic modifications of this rule which do lead to stable dynamics.

More specifically-- can the reduced, mean-field, description of the network+plasticity rule be leveraged to understand under what circumstances a Hebbian EE STDP rule could be stabilized? Alternatively, are there families of plasticity rules which are guaranteed to not stabilize the dynamics?

Does EE STDP combined with a specific form of inhibitory plasticity to do this job? (similar to the numerical work in Litwin Kumar Doiron 2014 which is mentioned in the discussion)?

Kohonen's rule that the authors investigate is stable-- but I am not aware of this rule having an electrophysiological basis. The authors do not cite literature to that effect, and so I do not agree that the current results support their statement (l. 440): "Therefore, this framework provides mathematical tractability and biological relevance to a model of excitatory STDP [25], without the need for unrealistically fast inhibitory STDP."

Other concerns and corrections

108 "only pairwise interactions are meaningful". Please clarify what "meaningful" means here, since a few lines afterwards the authors say they will consider more general examples in the future (115 "the BCM rule... depend on interactions beyond second order ..."). Another class of plasticity rules that cannot immediately be written in terms of the current theory are calcium based rules (Shouval et al 2002, Graupner Brunel 2012)

124 Timescale assumption.

The separation of timescales assumption should be introduced more clearly, on a number of levels:

1. Some readers may appreciate being given approximate values for the different timescales in the theory, perhaps in relation to the specific examples in Table 1.

2. The authors should discuss whether these assumptions are consistent with available data. For example Petersen, Malenka, ... Hopfield 1998, Froemke, Tsay, et al 2006, where ~10 repeats of a plasticity protocol are sufficient to induce significant changes or even saturation.

3. some computational studies indicate that the separation of timescales assumption must be violated for network stability, at least in relation to homeostatic plasticity (Zenke Gerstner Ganguli 2017). Does the theory provide a path for getting around these issues?

154 Intuitively I expect Neumann's approximation to break down for sparse matrices because the inverse \\bar{W}_0^{-1} is not well defined when the rank is not full. Is the 10% connectivity sufficiently large to avoid this issue? If yes, how small can the connection probability be before the approximation (and consequently the theoretical analysis) breaks down? Guzman, Schlogl, et al 2016 showed ~1% connectivity in hippocampal region CA3-- a region where recurrent connectivity and plasticity are thought to play important roles. The authors should comment on whether they expect their theory to be applicable in such a scenario.

158 \\Delta W can be defined such that the perturbation is done separately on EE, EI, IE, II connections. Does doing so within the current theory lead to the same results as Mongillo, Rumpel, Loewenstein 2018?

170 The writing in the first results paragraph should be improved to clarify this is an outline of the rest of the paper. Currently the different sections and applications of the theory come as a surprise. Otherwise I like the structure of the paper where introduction is mixed with development of the theory.

212 How does the stability depend on beta?

Fig. 1 B, C: are these rasters of only E neurons? what about I?

Fig. 1 D-I: what do the distributions of weights/rates/covariances look like? Kohonen's rule has the advantage that synaptic weights do not have a hard upper bound. Rather, \\beta plays the role of a soft upper bound. So the networks in these simulations have the potential of maintaining a realistic unimodal synaptic weight distribution, as seen in experiments, in contrast to previous theoretical studies where synaptic weights quickly go to their upper and lower bound (e.g. Rubin Lee Sompolinsky 2001). Is this indeed the case in these simulations?

The authors should indicate and discuss whether their simulations capture the corresponding experimentally observed properties of plastic neuronal networks beyond the averages.

Fig. 2 A, D: It seems that these plots can be used to estimate the characteristic time it takes the synaptic weights to reach steady state values. I think the authors should expand the discussion of these results (l. 232-241). What is the relationship between the slope of dJ/dt and \\tau_{STDP}? in other words, how many plasticity time constants does the synaptic weight transient last?

There is a very strong slowing down as a function of input correlations (panel D). Does this mean that if rates are small, we should expect synaptic weights to always be away from steady state for correlated inputs? If not, can an experimental prediction be extracted from this or will the transient timescale always be swamped by firing rates?

Fig. 2 C: is red E or I?

Fig. 3 A: Is it possible to add a line to this panel showing the actual average synaptic weight in simulations in a way that takes into account the initial conditions? I would expect such a line to be close to 0 for a range of j values, perhaps making the result in panel B easier to understand and less confusing after looking at panel A.

Fig. 3: increase font size of legend and inset figure labels.

257: The authors should cite Vegue Roxin 2019 here. They did take some steps in extending the MFT to cases with heterogeneous connectivity which leads to changes in the firing rate distribution (which are predictable given the weights).

304: From a theoretical point of view, it is nice that introduction of II plasticity (of the same form as EI) eliminates the heterogeneity of inhibitory firing rates. However, experimental data (e.g. Xue et al 2014, Chiu ,... Higley 2018) suggests that different inhibitory subtypes are subject to different inhibitory plasticity rules. I think it's fine to include this "trick" in a theoretical paper but I would at least mention the fact that inhibitory plasticity rules themselves are heterogeneous so in a real network we should not expect inhibitory firing rates to become similar due to homogeneity of the inhibitory plasticity rule.

331: missing citation

416: not sure that incapacity is a word. Consider using inability.

416: typo and missing ref. "I Ebsch ..."

426: The paragraph citing Ocker [57] should also cite Ravid-Tannenbaum Burak 2017

Appendix 1, Generating correlated spike trains.

I was not aware of this method. Its explanation is clear, but I lack intuition whether/why it is favorable over an alternative which seems simpler:

Generating a Gaussian process with the exact desired correlations/covariances

Computing the Gaussian CDF of the random number for each neuron at each time.

Computing the inverse Poisson CDF of the result.

(see a more complete description of the method in Macke Berens ... Bethge 2009)

Appendix 3, I think the authors should expand Appendix 3 to address my major concern 1: Are there minimal modifications of the classical EE STDP rule which imply a non-trivial solution to the fixed point equation (Eq. 2 in Appendix 3)?

Reviewer #2: The authors developed a self-consistent theory of rates, covariance and synaptic weights in balanced recurrent network that receives correlated external input. The theory can help to understand coexistence of correlated activity and dynamic excitatory-inhibitory balance. Furthermore, they found a plasticity rule for inhibitory-to-excitatory connections which have interesting property of dependence on initial condition. Finally, they showed that their theory could be applied to “out of balance” scenario of optogenetic input. Although, these are interesting and important claims we have some reservation about the amount of proof they provide for such a general claims and consequently about biological relevance of the study.

Major comment 1:

First claim is that authors developed a general theory of plasticity in balanced network which can address network dynamics with different plasticity rules and describe weight evolution with correlated activity. We have some concerns about generality, novelty and predictive power of the theory.

The theory presented is far from being a general theory of plasticity in balance networks. The claim hold and have be demonstrated by authors fairly well only in the case of tightly balanced large neural network (with assymptotic 1/sqrt(N) scaling) for the class of pairwise STDP rules (ignoring other plasticity types i.e. bcm, metaplasticity, intrinsic plasticity and structural plasticity). This STDP class authors chose is one with multiplicative dependence on synaptic weights which naturally lead to “nice” fixed points and for which mean-field description works very well.

The main theoretical contribution of the paper is procedure to put together and solve self-consistently equations for rate, covariance and weights. This theory seems very computational efficient and simple, but it is not the only one which can threat this problem. Equations 1 and 2 for rate and covariance with tight balance condition and large networks are already derived earlier (partly by the same authors Baker et al 2019 "Correlated states in balanced neuronal networks"), and there is even a statistical field theory developments in this area e.g. Ocker et al 2016 paper "Linking structure and activity in nonlinear spiking networks". The equation for individual weight evolution with input correlation were derived already by Gutig et al 2003 in paper "Learning Input Correlations through Nonlinear Temporally Asymmetric Hebbian Plasticity". The self-consistent weight evolution was done by Ocker et al 2015 in the paper "Self-Organization of Microcircuits in Networks of Spiking Neurons with Plastic Synapses", for the case of pairwise STDP rule which allows for weight heterogeneity and structure in connectivity (they chose a rule which is very sensitive to correlation in inputs). There is also work with higher-order SPDP rule, i.e. Monangie et al 2020 paper "Autonomous emergence of connectivity assemblies via spike triplet interactions".

To my best of my knowledge this is the first theoretical study of effect of external input correlation on tightly-balance balanced plastic networks. Similar theoretical attempts have been done before for the cases of external and self-generated correlations, e.g. Gilson et al 2009 “Emergence of network structure due to spike-timing-dependent plasticity in recurrent neuronal networks. I. Input selectivity–strengthening correlated input pathways” and Gilson 2010 "STDP in recurrent neuronal networks". Above studies did not have condition for tight balance and this work is based on this assumption. It is not clear if prescribing a balance a priory could bring clarity to the issue, because temporal dynamics of correlation can play a big role in unstable dynamical states, e.g. trough large Fano factor.

Major comment 2:

Second claim is that that correlations in the input mildly, but significantly affect the evolution of synaptic weights. This claim is well demonstrated, but it is a hardly surprising, given the type of the multiplicative pairwise rule authors use.

Major comment 3:

Third claim is that their inhibitory-to-excitatory plasticity induce correlations between activity and synaptic weights, such that final weight depends on initial synaptic weight. This is certainly interesting result, and especially because it could explain one role of inhibitory-to-inhibitory plasticity. That being said the rule used is very peculiar. The RHS of the ODE is scaled with initial position and authors did not fully clarify what is the consequence of this choice? What is biological justification? It is very confusing for reader that initial position is a parameter of an ODE. We would be interested to see a comparison of this rule to the original Vogels at al 2011 "Inhibitory plasticity balances excitation and inhibition in sensory pathways and memory networks" to properly understand the scaling choice. More importantly temporal evolution of the weights is not shown in this case.

Major comment 4:

Final claim is that this framework could be used to treat case of optogenetic input to subpopulation. We think that this claim is shown well, but it is the question if the study brings more clarity to the field than original Vogels 2011 study.

Reviewer #3: The manuscript addresses the dynamics of plasticity under STDP in randomly connected neural networks in the balanced regime. The manuscript provides some analytical insights into this topic, backed up by numerical simulations. The topic is important, and results seem substantial and interesting. Nevertheless, there are some significant weaknesses in the analysis and presentation, which I recommend to address before the manuscript is considered again for publication.

MAIN COMMENTS

Since the theory is derived at the level of mean population activities, there is a hidden assumption that synaptic weights evolve uniformly. Is this true empirically in the simulations? How variable are the weights following plasticity?

Related to this point, there is also an implicit assumption that weights evolve independently of other quantities. One example is the correlation between presynaptic firing rate and the synaptic weight, which is discussed in the section on inhibitory plasticity. But there are other possible correlations that could develop between synaptic strength and, for example: the in-degree and out-degree of involved neurons; under some rules competition between synapses belonging to the same neuron could also break some of the assumptions of the mean field theory.

A full theory of the plasticity dynamics should address these questions, but I accept that there is value in deriving the predictions of a mean field theory that ignores such correlations, while comparing the predictions with numerical results. The presentation should acknowledge these limitations more explicitly. The appearance of a discussion of correlations between firing rates and synapses only in the section on inhibitory plasticity is unsatisfactory.

Is there a way to understand why correlations did not play an important role in earlier sections?

While the theory is derived for dynamics of the synaptic weights, the numerical analysis does not demonstrate that the dynamics indeed follow equation 6. Instead, simulations are used only to demonstrate agreement with the predictions on the steady state. Do the dynamics follow the prediction?

I would also like to see some numerical support for statements that the network remains in an asynchrounous state following plasticity.

Lines 359-365: What happens under the perturbation without plasticity? How does this depend on parameters? How can we see that the network is in a balanced state following plasticity (but possibly not before plasticity)?

MINOR COMMENTS

In comparisons between simulations and the theory, for how long were the simulations run? What was the criterion used to identify that the weights reached steady state?

Line 6: it will be appropriate to cite also Shaham and Burak (2017) and Darshan et al (2017) in the context of spatially correlated activity in the balanced state.

In the discussion, it will be appropriate to cite Ravid-Tannenbaum and Burak (2016) in the context of the relationship between covariances and STDP dynamics.

Lines 58-59: why is the assumption not essential? How does this agree with Ref. 34, which shows that heterogeneity in the connection probabilities can destroy the balanced state?

PRESENTATION

It is inconvenient that only some of the equations are numbered. This makes it difficult to refer to the unnumbered equations when discussion the manuscript.

Equation 2: please define Gamma precisely, not only in limits, and either derive this result or provide a clear path from the results of previous papers to this expression, possibly in an appendix.

The role of Twin in equation 2 is not sufficiently clear. If I understood this correctly, Twin should be introduced already when defining the spike count covariance.

Lines 73-74: the statement that the second term in Equation 2 accounts for intrinsically generated variability is a bit confusing since the elements in the diagonal are negative. It will be helpful to clarify this.

The logic of how x and S combine to generate the STDP rule is mathematically simple, yet for many readers it will be difficult to understand the structure of the rule from the equations. It will be helpful to provide a graphical representation of the STDP rules in the different cases listed in Table 1.

Lines 140-148: I didn’t understand why root finding is involved in the numerical solution of the differential equations. Doesn’t equation 6 simply provide the required update of J, based on the values of r anc C that are obtained from equations 1-2?

Figure 3: “and the system predicts a stable fixed point”. Should “system” be replaced by “theory”?

Line 339: sentence ends prematurely.

The last equation in page 1 of SI1 includes parameters whose meaning is not defined.

In line 26 of SI1, units for dt are missing.

In line 53 of SI1, in the equation rm = r_x/cx, should cx be c_x? (where _ stands for subscript)

Lines 53-56: It is not easy to understand from the description how the daughter process spikes are obtained. I suggest to clarify that each daughter process is obtained by first taking all the spikes of the mother process, followed by a deletion of spikes with probability 1-c_x, and to mention explicitly that deletion is performed independently in each one of the daughter processes.

It will be more convenient to group all the supporting text in one file with several sections.

**Have all data underlying the figures and results presented in the manuscript been provided?**

Reviewer #1: Yes

Reviewer #2: Yes

Reviewer #3: None

PLOS authors have the option to publish the peer review history of their article (what does this mean?). If published, this will include your full peer review and any attached files.

Reviewer #1: No

Reviewer #2: No

Reviewer #3: No
---

## [Decision Letter · Decision Letter 1]

30 Dec 2020

Dear Dr. Josić,

Thank you very much for submitting your manuscript "Balanced Networks under Spike-Time Dependent Plasticity" for consideration at PLOS Computational Biology.

As with all papers reviewed by the journal, your manuscript was reviewed by members of the editorial board and by several independent reviewers. In light of the reviews (below this email), we would like to invite the resubmission of a significantly-revised version that takes into account the reviewers' comments.

As you can see, the three reviewers are mostly positive about your manuscript and find that it has improved. However, one of the reviewers still has the concern that the approach taken in this work is weaker than in previous works (especially the assumptions of infinite size networks and precise balance, which are more limiting that the assumptions made in other works). Therefore, the authors should make a clear case for the benefits of their approach.

In particular:

- Although the Introduction has improved in terms of putting this manuscript into the context of other published work, the link between descriptions of previous work and the goal of the current work is missing. This could be helped by elaborating more on existing work -- beyond just listing specific conditions under which previous frameworks hold (see also next comment).

- There are multiple references missing, some are incorrectly cited, and some equation numbers do not appear correctly (see reviewer C comments). For example, the authors write in their response letter that they cited Montangie et al. in the Introduction but they do not. It’s certainly more relevant than the studies of triplet STDP in feedforward networks with a single postsynaptic neuron that the authors currently cite, because it focuses on the plasticity of recurrent networks. It could and should be mentioned also in the Discussion where they discuss how their work can be extended to include the impact of second and higher order moments on the evolution of the weights. Another example is the Ravid-Tannenbaum and Burak (2016) reference which similarly to Montangie et al (2020) considers correlations generated intrinsically within the network -- that other works (e.g. Gilson et al.) do not. These are just some examples of how the authors could better explain how their work fits in the context of published work.

- Regarding notation and referencing to equations and figures, it might be a good idea to ensure that the entire paper is consistent as new things are added during the revision.

- Rather than stating all the results at the start of the Results section, the authors should work on providing a map of what they do (and why), rather than what they find. The specifics of what they find are best left to the individual sections where the specific figures are discussed and are currently confusing for such an introductory paragraph. I realize this was done in request of reviewer A, but I believe the authors were asked to clarify/outline their approach and goals (rather than state the results before actually presenting them).

- The authors highlighted parts of the paper in blue to indicate that they made changes but often times, only a single sentence in a paragraph is changed, or the order of two words is swapped. This leaves the false impression that many significant changes were made, when in fact, very few, if any changes were made. A clear example is the first paragraph of the Results, where only the first sentence was added and yet the entire paragraph is highlighted in blue. There are many such instances in the paper. It would be better and much clearer for the reviewers and editor to have only specific sentences (or words) that were changed highlighted, as the authors do for example near lines 326-328 and for the remainder of that section.

Minor:

- What is meant here: "Analytical treatments have been proposed for a number of cases, initially describing the impact of STDP on individual synaptic weights (Guettig)?” Do the authors mean in feedforward networks, or under some other special condition? Because any plasticity rule describes the impact of STDP on individual synaptic weights. This comment is related to the first comment above, namely to be clear about what other works did, and how this work is different from them.

- Shouldn’t the Results section where the authors introduce the Kohonen rule (around line 221) have a subsection titled something like “Dynamics of balanced networks under the Kohonen rule” to match the corresponding section starting in line 252?

- Some redundancies (there are probably others) that should be eliminated to make the narrative clear:

e.g. (1) line 218: “Our theory explains the effect of such plasticity rules in balanced networks” and line 220: “the impact of the plasticity rule on the dynamics of the network”

(2) lines 37 and 47: “how synaptic plasticity impacts (shapes) network dynamics”

(3) lines 131 and 137: “focus on pairwise interactions” vs “considered interaction up to second order”

We cannot make any decision about publication until we have seen the revised manuscript and your response to the reviewers' comments. Your revised manuscript is also likely to be sent to reviewers for further evaluation.

Sincerely,

Julijana Gjorgjieva, PhD

Guest Editor

PLOS Computational Biology

Kim Blackwell

Deputy Editor

PLOS Computational Biology

As you can see, the three reviewers are mostly positive about your manuscript and find that it has improved. However, one of the reviewers still has the concern that the approach taken in this work is weaker than in previous works (especially the assumptions of infinite size networks and precise balance, which are more limiting that the assumptions made in other works). Therefore, the authors should make a clear case for the benefits of their approach.

In particular:

- Although the Introduction has improved in terms of putting this manuscript into the context of other published work, the link between descriptions of previous work and the goal of the current work is missing. This could be helped by elaborating more on existing work -- beyond just listing specific conditions under which previous frameworks hold (see also next comment).

- There are multiple references missing, some are incorrectly cited, and some equation numbers do not appear correctly (see reviewer C comments). For example, the authors write in their response letter that they cited Montangie et al. in the Introduction but they do not. It’s certainly more relevant than the studies of triplet STDP in feedforward networks with a single postsynaptic neuron that the authors currently cite, because it focuses on the plasticity of recurrent networks. It could and should be mentioned also in the Discussion where they discuss how their work can be extended to include the impact of second and higher order moments on the evolution of the weights. Another example is the Ravid-Tannenbaum and Burak (2016) reference which similarly to Montangie et al (2020) considers correlations generated intrinsically within the network -- that other works (e.g. Gilson et al.) do not. These are just some examples of how the authors could better explain how their work fits in the context of published work.

- Regarding notation and referencing to equations and figures, it might be a good idea to ensure that the entire paper is consistent as new things are added during the revision.

- Rather than stating all the results at the start of the Results section, the authors should work on providing a map of what they do (and why), rather than what they find. The specifics of what they find are best left to the individual sections where the specific figures are discussed and are currently confusing for such an introductory paragraph. I realize this was done in request of reviewer A, but I believe the authors were asked to clarify/outline their approach and goals (rather than state the results before actually presenting them).

- The authors highlighted parts of the paper in blue to indicate that they made changes but often times, only a single sentence in a paragraph is changed, or the order of two words is swapped. This leaves the false impression that many significant changes were made, when in fact, very few, if any changes were made. A clear example is the first paragraph of the Results, where only the first sentence was added and yet the entire paragraph is highlighted in blue. There are many such instances in the paper. It would be better and much clearer for the reviewers and editor to have only specific sentences (or words) that were changed highlighted, as the authors do for example near lines 326-328 and for the remainder of that section.

Minor:

- What is meant here: "Analytical treatments have been proposed for a number of cases, initially describing the impact of STDP on individual synaptic weights (Guettig)?” Do the authors mean in feedforward networks, or under some other special condition? Because any plasticity rule describes the impact of STDP on individual synaptic weights. This comment is related to the first comment above, namely to be clear about what other works did, and how this work is different from them.

- Shouldn’t the Results section where the authors introduce the Kohonen rule (around line 221) have a subsection titled something like “Dynamics of balanced networks under the Kohonen rule” to match the corresponding section starting in line 252?

- Some redundancies (there are probably others) that should be eliminated to make the narrative clear:

e.g. (1) line 218: “Our theory explains the effect of such plasticity rules in balanced networks” and line 220: “the impact of the plasticity rule on the dynamics of the network”

(2) lines 37 and 47: “how synaptic plasticity impacts (shapes) network dynamics”

(3) lines 131 and 137: “focus on pairwise interactions” vs “considered interaction up to second order”

Reviewer's Responses to Questions

**Comments to the Authors:**

Reviewer #1: The authors have addressed all my comments from the original review.

In the revised form the manuscript is an important contribution to the field, so support publishing it in PLoS CB.

Minor corrections:

Supporting figure 2 C

Please change the graphics to indicate more clearly that η = 1 is shown in the inset only, similar to panel B.

Supporting figure 3

Missing equation cross reference

Reviewer #2: I think the manuscript has been greatly improved by the authors and my technical concerns were fully answered. Therefore I have no issues with the presented theory which gives us an alternative way of calculating rate-correlation-weights self-consistency, but it is hard for me to see how this approach can be better than previous attempts. In my opinion authors still have to provide a case study to show that this approach could give us a new insight to an open problem or could do the same tasks as the previous models, but better (more efficiently or more correctly).

The reason I believe that this theory cannot do better than previous attempts is because it has a lot of constraints. The most problematic ones are the assumption of limit N tends to infinity, the assumption of precise balance and the assumption that there is a clear separation of time-scales. Those constraints could contradict the essential requirement of biological plausibility.

The theory does not capture dynamics of finite size networks, which are the case of biological neural networks. Does the author assume that the transfer function of single neurons don't play any role? In this respect I am yet to be convinced that this approach would be better then Ocker et al 2015. Especially for calculating rates and correlation self-consistently (Ocker et al. 2016). Is the assumption of infinite networks really weaker than the assumption on the structure of input noise and spike statistics? Until there is a good use case I am inclined to believe that this approach mostly ignores single neuron dynamics and it would be less useful.

Although biological networks keep the EI balance, there is a fundamental question on how do we model it? It is not a priori necessary to keep balance with 1/\\sqrt{K} scaling. This is the case for current based synapses and this scaling was first introduced for the purpose of theoretical simplification. If we have more biological realistic neurons with conductance based synapses, the balance is kept dynamically. Furthermore, there are networks which are not necessarily balanced, e.g. in development or diseases. Therefore, I would put more utility in the approach which can treat the problem of self-consistency without the balance condition, but can handle a balanced network as a special case. On this point I would like authors to present a case where their framework works better than Burkit AN et al. 2007, Gilson et al. 2009 or Ocker et. al 2015 approach.

Reviewer #3: The revision addresses my comments, but in a few cases there are remaining minor issues that should be taken care of. Please see the attached file. My comments are posted over the pdf (five comments overall, in pages 26, 29, 30, 30, 32).

**Have all data underlying the figures and results presented in the manuscript been provided?**

Reviewer #1: Yes

Reviewer #2: Yes

Reviewer #3: Yes

PLOS authors have the option to publish the peer review history of their article (what does this mean?). If published, this will include your full peer review and any attached files.

Reviewer #1: No

Reviewer #2: No

Reviewer #3: No
---

## [Decision Letter · Decision Letter 2]

12 Apr 2021

Dear Dr. Josić,

We are pleased to inform you that your manuscript 'Balanced Networks under Spike-Time Dependent Plasticity' has been provisionally accepted for publication in PLOS Computational Biology. Please just remove/rephrase the term: "general theory of plasticity" in the abstract, introduction and elsewhere from the manuscript, as per the suggestion of Rev 2, because as you acknowledge, the proposed theory is only appropraite under certain assumptions.

Best regards,

Julijana Gjorgjieva, PhD

Guest Editor

PLOS Computational Biology

Kim Blackwell

Deputy Editor

PLOS Computational Biology

Reviewer's Responses to Questions

**Comments to the Authors:**

Reviewer #2: Thank you for further clarifying your work. I agree with the major points. Most importantly, that there is still no efficient theoretical approach involving Fokker-Planck with colored noise and strong input correlation, and certainly not one applied to solve rate-correlation-weight self-consistency, which makes your work more universally applicable.

Small remark.

I'm not sure the term general theory of plasticity is the best.

Reviewer #3: The revision addresses the comments raised in my previous review.

**Have the authors made all data and (if applicable) computational code underlying the findings in their manuscript fully available?**

Reviewer #2: None

Reviewer #3: None

PLOS authors have the option to publish the peer review history of their article (what does this mean?). If published, this will include your full peer review and any attached files.

Reviewer #2: No

Reviewer #3: No

---

## [Editor Report · Acceptance letter]

10 May 2021

PCOMPBIOL-D-20-00683R2 

Balanced Networks under Spike-Time Dependent Plasticity

Dear Dr Josić,

I am pleased to inform you that your manuscript has been formally accepted for publication in PLOS Computational Biology. Your manuscript is now with our production department and you will be notified of the publication date in due course.

With kind regards,

Katalin Szabo
